# Real-World Data Regarding Satisfaction to Botulinum Toxin A Injection into the Urethral Sphincter and Further Bladder Management for Voiding Dysfunction among Patients with Spinal Cord Injury and Voiding Dysfunction

**DOI:** 10.3390/toxins14010030

**Published:** 2022-01-02

**Authors:** Cheng-Ling Lee, Jia-Fong Jhang, Yuan-Hong Jiang, Hann-Chorng Kuo

**Affiliations:** Department of Urology, Hualien Tzu Chi Hospital, Buddhist Tzu Chi Medical Foundation, Tzu Chi University, Hualien 970, Taiwan; leecl@hotmail.com (C.-L.L.); alur1984@hotmail.com (J.-F.J.); redeemer1019@yahoo.com.tw (Y.-H.J.)

**Keywords:** neurogenic bladder, detrusor sphincter dyssynergia, botulinum toxin, urodynamic study

## Abstract

Purpose: This study aimed to investigate improvement in voiding condition after the initial botulinum toxin A (BoNT-A) injection into the urethral sphincter among patients with chronic spinal cord injury (SCI) and voiding dysfunction. Moreover, subsequent surgical procedures and bladder management were evaluated. Materials and Methods: From 2011 to 2020, 118 patients with SCI and dysuria who wanted to void spontaneously received their first BoNT-A injection at a dose of 100 U into the urethral sphincter. Improvement in voiding and bladder conditions after BoNT-A treatment were assessed. Next, patients were encouraged to continually receive BoNT-A injections into the urethral sphincter, convert to other bladder managements, or undergo surgery. After undergoing bladder management and surgical procedures, the patients were requested to report improvement in voiding condition and overall satisfaction to bladder conditions. Then, data were compared. Results: In total, 94 male and 24 female participants were included in this analysis. Among them, 51 presented with cervical, 43 with thoracic, and 24 with lumbosacral SCI. After BoNT-A injections into the urethral sphincter, 71 (60.2%) patients, including 18 (15.3%) with excellent, and 53 (44.9%) with moderate improvement, had significant improvement in voiding condition. Patients with cervical SCI (66.6%), detrusor overactivity and detrusor sphincter dyssynergia (72.0%), partial hand function (80.0%), and incomplete SCI (68.4%) had a better improvement rate than the other subgroups. Only 42 (35.6%) patients continually received treatment with BoNT-A injections into the urethral sphincter. Meanwhile, more than 60% of patients who converted their treatment to augmentation enterocystoplasty (*n* = 5), bladder outlet surgery (*n* = 25), BoNT-A injections into the detrusor muscle (*n* = 20), and medical treatment (*n* = 55) had moderate and marked improvement in voiding dysfunction and overall satisfaction. Discussion: Although BoNT-A injections into the urethral sphincter could improve voiding condition, only patients with SCI who presented with voiding dysfunction were commonly satisfied. Those whose treatments were converted to other bladder managements, which can promote urinary continence, or to surgical procedures, which can facilitate spontaneous voiding, had favorable treatment outcomes.

## 1. Introduction

Voiding dysfunction and urinary incontinence are common neurogenic lower urinary tract dysfunctions among patients with chronic spinal cord injury (SCI). Patients with SCI caused by suprasacral lesions frequently present with neurogenic detrusor overactivity (NDO) and detrusor sphincter dyssynergia (DSD) [1] and those with SCI above the T6 level might have autonomic dysreflexia (AD). Nearly half of patients with untreated DSD will develop deleterious urologic complications due to high intravesical pressures, resulting in urolithiasis, urinary tract infection (UTI), vesicoureteral reflux, hydronephrosis, obstructive uropathy, and renal failure. [2]. In the management of voiding dysfunction among patients with SCI who presented with DSD and/or AD, clean intermittent catheterization (CIC) is the primary choice if medical treatment and bladder triggering are not effective in evacuating the bladder adequately [3].

DSD is associated with a high risk of complications, such as UTI, obstructive uropathy, and hydronephrosis. Antimuscarinic medication and catheterization are the mainstay treatment. However, in patients who cannot receive such treatments, external sphincterotomy has been the conventional management [4]. However, this procedure is associated with significant risks, including hemorrhage, erectile dysfunction, and the need for repeat procedures. Over the last decade, other alternatives, such as urethral stents and botulinum toxin A (BoNT-A) injection have been investigated [5].

Initially, BoNT-A was used to treat patients with severe dysuria caused by SCI and DSD [6]. The paralytic effect of BoNT-A injections on the external urethral sphincter in patients with neurogenic voiding disorders was investigated prospectively. BoNT-A injections, which aim to suppress DSD but not bladder neck dyssynergia, can be a possible alternative among patients who do not want to undergo surgery or who cannot perform self-catheterization [7]. Reduced urethral sphincter hypertonicity via chemical denervation was noted after BoNT-A treatment, and patients can then urinate more efficiently. BoNT-A has been safely used for the treatment of neurogenic urethral sphincter spasticity among patients with DSD caused by SCI, Parkinson’s disease, cerebrovascular accident, and multiple sclerosis. The treatment outcome was initially satisfactory. However, adverse events including increased urinary incontinence and incomplete bladder emptying were observed [7].

BoNT-A injection into the urethral sphincter can decrease urethral pressure, post-void residual (PVR) volume, and AD, and can increase maximum flow rate (Qmax) in patients with DSD caused by SCI [6,7,8]. External sphincter hypertonicity might have different severity. Therefore, BoNT-A injection at a dose of 100 U might not be efficient in facilitating voiding among patients with high-grade DSD. In addition, patients with NDO and DSD have uninhibited DO. Thus, these patients might develop a higher urinary incontinence grade after BoNT-A injection into the urethral sphincter [9]. Therefore, patients with SCI who were initially treated with BoNT-A injection into the urethral sphincter and wanted to resume spontaneous voiding may not appreciate the treatment outcome, and may want convert to other bladder managements or surgical procedures to achieve a better voiding condition with more satisfactory outcomes.

There was a discrepancy between the objective urodynamic outcomes and patient satisfaction in the treatment of DSD with BoNT-A injection. Approximately 60.6% and 77.3% of patients with SCI and DSD who received urethral and detrusor BoNT-A injection, respectively, had satisfactory overall outcomes. The detrusor group had a significantly better improvement in Urogenial Distress Inventory-6 and International Impact Questionnaire short form-7 scores than the urethral injection group. A high incontinence grade was the major cause of dissatisfaction in the urethral injection group [10]. 

The current study aimed to investigate improvement in voiding condition after the initial BoNT-A injection into the urethral sphincter and subsequent surgical procedures and bladder management among patients with chronic SCI and voiding dysfunction. Moreover, the predictive factors for a satisfactory treatment outcome after urethral sphincter BoNT-A injection, and the overall satisfaction to surgical procedures and other bladder managements for DSD and voiding dysfunction among patients with SCI were assessed.

## 2. Results

In total, 118 patients were recruited during the 10-year study. Among them, 94 were male and 24 were female patients. Moreover, 51 presented with cervical, 43 with thoracic, and 24 with lumbosacral SCI. Table 1 shows the characteristics of patients. After BoNT-A injections into the urethral sphincter, 71 (60.2%) patients, including 18 (15.3%) with markedly and 53 (44.9%) with moderate improvement, had improvement in voiding condition. Patients with cervical SCI (66.6%), NDO + DSD (72.0%), partial hand function (80.0%), and incomplete SCI (68.4%) had a better improvement rate than those in the other subgroups. However, if we compared the rate of patients with moderately and markedly improvement among subgroups with different SCI level, videourodynamic characteristics, presence of AD, hand function and completeness of hand function, no significant difference was noted among subgroups.

Compared with the baseline urodynamic parameters between the subgroups with different improvement in voiding condition, a lower detrusor pressure at baseline was the only significant factor for predicting improvement. The other urodynamic parameters were not effective in predicting the therapeutic effect of urethral sphincter BoNT-A injection among patients with SCI who presented with voiding dysfunction. (Table 2).

After the initial BoNT-A injections into the urethral sphincter, 55 (46.6%) patients discontinued the treatment due to inefficiency or urinary incontinence exacerbation. Moreover, 42 (35.6%) patients continually received BoNT-A injections into the urethral sphincter every 6–12 months. In total, 20 (16.9%) converted to BoNT-A injection into the detrusor muscle and CIC. Surgical procedures were performed to facilitate spontaneous voiding or to restore urinary continence. After undergoing different surgical procedures and bladder management, improvement in bladder condition (frequency, urgency, and urinary continence) and voiding condition were assessed, and the results are shown in Table 3. Patients who continually received BoNT-A injections into the urethral sphincter did not experience significant improvement in bladder and voiding conditions (28.6%). However, patients who converted to other bladder management, which can promote urinary continence, or those who underwent surgery, which can facilitate spontaneous voiding, had favorable treatment outcomes.

After a median 5-year follow-up of bladder and voiding conditions after undergoing other bladder management and surgical procedures, patients with SCI reported overall satisfaction to their current bladder and voiding conditions. Only 40 (33.9%) patients were extremely satisfied, and 30 (25.4%) were satisfied but wanted to change their management to achieve a better voiding or continent bladder condition. Furthermore, 41 (41.7%) patients were unsatisfied but did not want to change their management, and 7 (5.9%) were extremely unsatisfied. The overall satisfaction rate of patients with a suburethral sling for urinary incontinence (100%), augmentation enterocystoplasty with CIC (80%), transurethral incision of the bladder neck, which can facilitate spontaneous voiding (62.5%), and CIC alone (50%) was >50% (Table 4). If we combine the number of patients who were satisfied to their current bladder condition, the rate of satisfaction to most bladder managements and surgical procedures could be >60%. The most common causes of dissatisfaction were persistent voiding difficulty, urinary incontinence, small volume of voided urine, and need for CIC for bladder emptying.

## 3. Discussion

This retrospective study revealed that satisfaction to BoNT-A injection into the urethral sphincter was limited in most patients with chronic SCI who presented with voiding dysfunction. Only one-third of patients who continually received urethral BoNT-A injections experienced improvement in voiding condition. However, patients who converted to other bladder management, which can promote urinary continence, or who underwent surgical procedures, which can facilitate spontaneous voiding, had favorable treatment outcomes.

BoNT-A injection into the urethral sphincter was first used to treat DSD in patients with SCI to facilitate spontaneous urination without self-catheterization [8]. The therapeutic effect commonly appears 1 week after injection, and the treatment efficacy lasts for 3–6 months [7]. The common dose of BoNT-A injection into the urethral sphincter for DSD is 100 U [11,12,13]. After BoNT-A injection, the PVR volume and maximal urethral closure pressure decreased, and the Qmax increased [7]. BoNT-A injection into the urethral sphincter can be performed transperineally or transurethrally, and previous studies showed that such a treatment was effective [14]. The incidence of UTI among patients with NDO reduced by 50% after BoNT-A injection into the urethral sphincter, mainly due to a lower PVR and intravesical pressure after treatment [13]. However, BoNT-A injection into the urethral sphincter is not widely applied on patients with SCI because of adverse events including increased urinary incontinence and persistent incomplete bladder emptying after injection [15,16,17]. Exacerbation of urinary incontinence among patients with SCI commonly prohibits the clinical use of BoNT-A injection into the urethral sphincter in treating DSD, and patients want switch to BoNT-A injection into the detrusor muscle to achieve urinary continence even though CIC is required [10,15,18].

Patients with SCI who presented with DSD do not commonly prefer BoNT-A injection into the urethral sphincter. However, patients with neurogenic lesion and discoordinated urethral sphincter during urination, and those who are ambulatory and prefer to urinate voluntarily are likely to benefit from this treatment [19]. Previous studies revealed that BoNT-A injection into the urethral sphincter at a dose of 100–200 U was effective in patients with multiple sclerosis, cerebrovascular accident, or SCI [11,16]. CIC can be prevented in patients with cerebrovascular accident and chronic urinary retention after BoNT-A injection into the urethral sphincter at a dose of 100 U [20]. The duration of treatment efficacy is between 2 and 9 months based on the number of injections. However, repeat injection is required to maintain therapeutic efficacy [21]. Moreover, patients with SCI might convert to other bladder management or undergo surgical procedures to achieve a more definite treatment outcome without subsequent repeat treatments [15]. This study showed that patients with cervical SCI, partial hand function, and incomplete SCI lesion had a more favorable improvement in voiding condition. Thus, patients with SCI who can handle bladder management by themselves will more likely accept this treatment modality.

Based on baseline urodynamic parameters, patients with significant improvement in voiding condition had a lower detrusor pressure at baseline, thereby indicating that less tightness in the urethral sphincter can be a possible predictor of favorable treatment outcome after BoNT-A injection into the urethral sphincter. A previous study showed that patients with severe spasticity over the external urethral sphincter may require repeated injections or higher doses of BoNT-A. Patients with good treatment outcomes had a significantly lower baseline integrated electromyography [22]. Even after repeated urethral sphincter BoNT-A injection, patients did not report a high improvement rate in this study.

The poor treatment outcome of BoNT-A injection into the urethral sphincter is attributed to not only a more severe DSD but also the presence of bladder neck dyssynergia and the absence of detrusor contraction [23]. This research revealed that patients with DU and DSD had less favorable treatment outcomes after the initial BoNT-A injections into the urethral sphincter. Meanwhile, patients who converted to bladder outlet surgery such as transurethral incision of the bladder neck, transurethral incision of the prostate, and transurethral resection of the prostate had a higher rate of improvement in voiding condition. Interestingly, patients who continually received BoNT-A injection into the urethral sphincter did not experience any improvement in voiding dysfunction. The treatment outcome of voiding condition did not meet their expectation after repeated BoNT-A injections into the urethral sphincter. This treatment modality when used on all patients with DSD does not have convincing results, and it should only be performed in specific patients with chronic SCI. Dissatisfaction was mainly attributed to a high incontinence grade that was not anticipated before urethral BoNT-A injection. Less difficulty in urination and less PVR requiring CIC were the major causes of satisfaction. Meanwhile, an increase in incontinence grade and the need for CIC were mainly associated with dissatisfaction among patients receiving BoNT-A injection into the urethral sphincter [15].

Although BoNT-A injection into the urethral sphincter is effective in 60% of patients with voiding dysfunction refractory to conventional medical treatment, urinary incontinence can be a possible de novo adverse event after such a therapy. Patients might be disappointed due to the presence of urinary incontinence that might exceed voiding difficulty after BoNT-A injection into the urethral sphincter. Therefore, patients with SCI who presented with voiding dysfunction should be informed about its limited therapeutic efficacy against voiding dysfunction and possible adverse event of urinary incontinence before treatment [24].

This study showed a discrepancy between self-reported improvement in voiding condition and overall satisfaction. Patients with DSD commonly present with bladder storage and emptying symptoms. Although patients did experience improvement in voiding condition after BoNT-A injection into the urethral sphincter, they were still satisfied with their bladder condition due to the absence of urinary incontinence. Not all patients who underwent bladder outlet surgery could experience significant improvement in voiding dysfunction because a tight external sphincter remains a functional obstruction during voiding. Patients had satisfactory voiding or bladder condition after surgery or further bladder management. However, a high percentage of patients with SCI want to change their management to achieve a more favorable outcome, which is adequate self-voiding without urinary incontinence. However, this might be an unrealistic expectation.

In addition to voiding via abdominal tapping, CIC must be performed by patients with DSD themselves or caregivers. However, some patients with DSD prefer spontaneous voiding without performing CIC, and they might want to be dry after treatment even if CIC is necessary. Therefore, the management of voiding dysfunction and incontinence among patients with SCI and DSD is a challenge for physicians, and should be considered as an art. Notably, the overall satisfaction (very satisfied and satisfied but wish to change) to BoNT-A injection into the detrusor muscle and augmentation enterocystoplasty was not inferior to that after transurethral incision of the prostate, and transurethral resection of the prostate, and transurethral incision of the bladder neck. Therefore, patients who became dry after bladder management with CIC had a high satisfaction rate than those who could void spontaneously after surgery.

Approximately 95% of patients with suprasacral lesions presented with detrusor overactivity with or without DSD [25]. Hand dexterity, abdominal muscle power, bladder sensation, and degree of urethral sphincter DSD might affect VE and neurogenic lower urinary tract dysfunction. Although urethral BoN-A injections can reduce urethral resistance and facilitate spontaneous voiding via percussion, de novo urinary incontinence, incomplete bladder emptying, and subsequent UTI can still be bothersome for patients with SCI [11,26]. Combined detrusor and urethral BoNT-A injections might achieve the desired goals, including decreased urinary incontinence and improved voiding efficiency [19]. Hence, patients must be educated regarding the acceptance of their current condition and bladder management with the least invasive procedure without further surgery, and these actions might be reasonable. If patients with SCI can accept their current condition and choose the most suitable bladder management and minimally invasive procedure, satisfaction to voiding dysfunction or bladder management will overweigh their expectation of spontaneous voiding.

## 4. Conclusions

The therapeutic efficacy of BoNT-A injection into the urethral sphincter for voiding dysfunction in patients with chronic SCI is limited. The rate of improvement in voiding dysfunction after repeated BoNT-A injection into the urethral sphincter was less favorable than that after surgery or medical treatment. Nevertheless, patients with SCI commonly accept their voiding condition, and they were satisfied with the treatment outcome.

## 5. Materials and Methods

This retrospective study analyzed consecutive patients with SCI with voiding dysfunction who were refractory to medical treatment and who received BoNT-A injection (onabotulinumtoxinA, Allergan, Irvine, CA, USA) at a dose of 100 U into the urethral sphincter from 2011 to 2020. The data of patients in this study were collected from one medical center and all treatments were performed by the same urology team. All patients underwent video urodynamic study to identify the underlying pathophysiology of neurogenic lower urinary tract dysfunction before BoNT-A injection into the urethral sphincter. Only patients with DSD who wanted to achieve spontaneous voiding via percussion or abdominal straining were included in the final analysis. All patients were informed about the possible adverse events after BoNT-A injection into the urethral sphincter before treatment. This study had been approved by the Research Ethics Committee of Hualien Tzu Chi Hospital (IRB: 110-033-B). Given the retrospective nature of this study, the requirement for informed consent was waived by the Research Ethics Committee of Hualien Tzu Chi Hospital. All methods used in this study were conducted in accordance with relevant guidelines and regulations.

All patients underwent video urodynamic study at baseline. The video urodynamic study parameters included first bladder sensation of filling, cystometric bladder capacity, Qmax, detrusor pressure at Qmax, PVR, and intra-abdominal pressure on voiding in patients with detrusor underactivity. Moreover, voiding efficiency was recorded. The terminology used in this study was based on the recommendations of the International Continence Society [27]. To analyze treatment outcome, voiding dysfunctions were categorized into NDO without DSD (NDO–DSD), NDO + DSD, NDO + AD + DSD, DU, and intrinsic sphincter deficiency, according to the filling phase of video urodynamic study and the voiding phase of pressure flow study. Moreover, electromyographic characteristics and images during the voiding phase were analyzed.

BoNT-A injection was performed in the operating room under light intravenous general anesthesia. In total, a dose of 100 U was administered on the transurethral sphincter [11]. One vial of 100 U onabotulinumtoxinA was reconstituted with 4 mL of normal saline, thereby achieving a concentration equivalent to 25 U per mL. In total, 1 mL of BoNT-A solution was injected into the urethral sphincter transurethrally at 3-, 6-, 9-, and 12 o’clock positions among men, and into the urethral sphincter along the urethral lumen transcutaneously at 1-, 4-,7-, and 10-o’clock positions at the urethral meatus side among women.

A Foley catheter was inserted overnight after BoNT-A injections, and patients were instructed to report their voiding condition at the outpatient clinic after 1 month. The effect of BoNT-A injection on urethral sphincter function was observed 2–3 days after treatment. The maximum therapeutic effect could reach up to 2 weeks after BoNT-A injection [11]. Patients with detrusor underactivity and a large PVR were instructed to void using the Crede maneuver or abdominal straining, and CIC, instead of an indwelling Foley catheter, was recommended. Patients were recommended to perform CIC even if they could urinate via percussion, triggering, or application of abdominal pressure. The PVR was assessed, and CIC was continued until the PVR decreased to <50% of the voided volume. Antibiotics were routinely administered for 3 days to prevent UTI after BoNT-A injections into the urethral sphincter.

The treatment outcome was assessed based on self-reported data about improvement in voiding condition and overall satisfaction to the current bladder condition after BoNT-A injection into the urethral sphincter. Patients were instructed to report improvement in voiding condition as no improvement (no change in voiding condition after BoNT-A injection), mild improvement (less difficulty in urination and a decrease in CIC frequency to <50%, which similar to the preoperative value), moderate improvement (without difficulty in urination and a decrease in CIC frequency to <25%, which is similar to the preoperative value), significant improvement (without difficulty in urination and CIC was not necessary, according to their subjective perception of ease to urination via percussion or abdominal pressure after BoNT-A injection into the urethral sphincter).

Patients were encouraged to continue BoNT-A injections into the urethral sphincter if they had moderate to significant improvement in voiding condition. If there was no therapeutic effect, patients converted to other bladder managements (including medical treatment, CIC, and indwelling Foley catheter), or underwent surgical procedure and bladder management according to the baseline video urodynamic study findings. The surgical procedures included cystostomy diversion, detrusor BoNT-A injection and CIC, augmentation enterocystoplasty and CIC, transurethral incision of the bladder neck, transurethral incision of the prostate, transurethral resection of the prostate, suburethral sling for intrinsic sphincter deficiency, and external sphincterotomy (Figure 1). After undergoing bladder management and surgical procedures, patients were instructed to report improvement in voiding condition and overall satisfaction to their current bladder and voiding conditions (very satisfied, satisfied but need change, unsatisfied but no change, and very unsatisfied).

Continuous variables were expressed as means ± standard deviations and categorical data as numbers and percentages (%). Statistical analysis was performed to compare the patient subgroups with moderately and markedly improved treatment outcomes; the rates of patients with moderately and markedly improved treatment outcomes after each bladder management or surgical procedure; and the rates of very satisfied outcome and satisfied but wish to change versus unsatisfied but no change and very unsatisfied outcome after each bladder management or surgical procedure. To determine the *p*-values between subgroups for statistical comparisons, the chi-square test was used for categorical variables and the Wilcoxon signed-rank test for continuous variables. All statistical assessments were two-sided, and a *p* value of <0.05 was considered significant. All calculations were performed using the Statistical Package for the Social Sciences software for Windows version 16.0 (SPSS, Chicago, IL, USA).

## Figures and Tables

**Figure 1 toxins-14-00030-f001:**
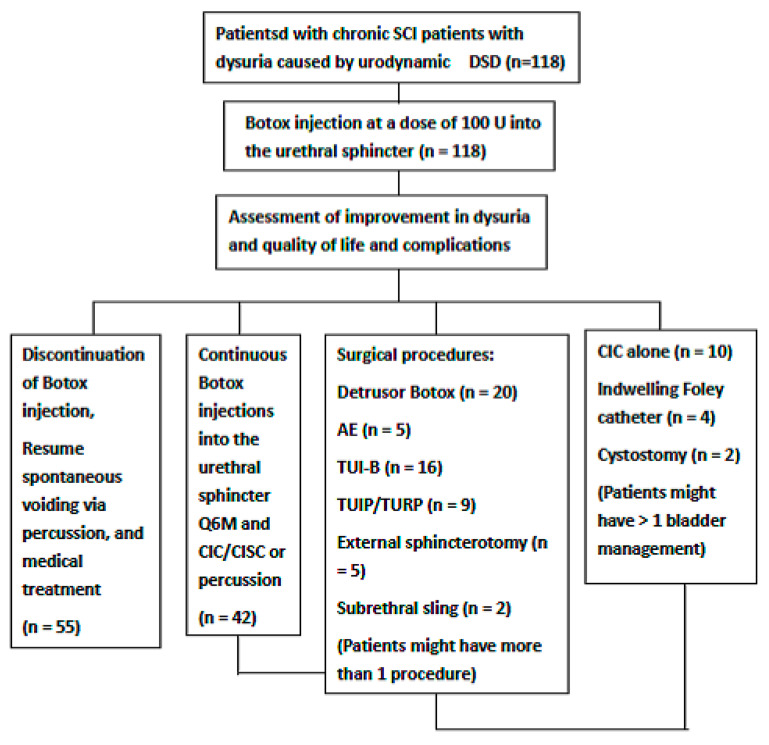
Flow chart of the surgical procedure and bladder management after the initial botulinum toxin A injection into the urethral sphincter among patients with spinal cord injury and detrusor sphincter dyssynergia.

**Table 1 toxins-14-00030-t001:** Improvement in voiding condition according to the characteristics of patients with spinal cord injury and detrusor sphincter dyssynergia who received botulinum toxin A (BoNT-A) injection into the urethral sphincter.

	N	Not Improved	Mildly Improved	Moderately Improved	Markedly Improved	*p* Value *
Total	118	8 (6.8%)	39 (33.1%)	53 (44.9%)	18(15.3%)	
Male Female	94 24	8 (8.5%) 0	31 (33.0%) 8 (33.3%)	39 (41.5%) 14 (58.3%)	16 (17.0%) 2 (8.3%)	0.288
Level of SCI Cervical Thoracic Lumbo-sacral	51 43 24	4 (7.8%) 3 (7.0%) 1 (4.2%)	13 (25.5%) 16 (37.2%) 10 (41.7%)	25 (49.0%) 18 (41.9%) 10 (41.7%)	9 (17.6%) 6 (14.0%) 3 (12.5%)	0.866
NDO-DSD NDO + AD + DSD NDO + DSD DA + DSD DA+ ISD	5 57 25 30 1	0 4 (7.0%) 1 (4.0%) 3 (10.0%) 0	3 (60.0%) 16 (28.1%) 6 (24.0%) 14 (46.7%) 0	0 32 (56.1%) 11 (44.0%) 9 (30.0%) 1 (100%)	2 (40.0%) 5 (8.8%) 7 (28.0%) 4 (13.3%) 0	0.052
With AD Non-AD	87 31	5 (5.7%) 3 (9.7%)	31 (35.6%) 8 (25.8%)	41 (47.1%) 12 (38.7%)	10 (11.5%) 8 (25.8%)	0.190
Hand function Normal Partial Incapable	89 15 14	6 (6.7%) 0 2 (14.3%)	33 (37.1%) 3 (20.0%) 3 (21.4%)	40 (44.9%) 7 (46.7%) 6 (42.9%)	10 (11.2%) 5 (33.3%) 3 (21.4%)	0.199
Complete Incomplete	80 38	7 (8.8%) 1 (2.6%)	28 (35.0%) 11 (28.9%)	33 (41.3%) 20 (52.6%)	12 (15.0%) 6 (15.8%)	0.482

* *p* value: statistical analysis of comparison of the rate of patients with moderately and markedly improvement among different subgroups of patient and videourodynamic characteristics, SCI: spinal cord injury, NDO: neurogenic detrusor overactivity, DSD: detrusor sphincter dyssynergia, AD: autonomic dysreflexia, ISD: intrinsic sphincter deficiency.

**Table 2 toxins-14-00030-t002:** Baseline urodynamic parameters of patients with improvement in voiding condition at varying degrees after the initial BoNT-A injection into the urethral sphincter.

Urodynamic Parameters	Not Improved	Mildly Improved	Moderately Improved	Markedly Improved	*p* Value
Patient no.	8 (6.8%)	39 (33.1%)	53 (44.9%)	18 (15.3%)	
Age	49.2 ± 17.3	48.2 ± 14.8	46.5 ± 18.4	50.4 ± 15.2	0.842
Duration	6.8 ± 8.0	8.3 ± 11.1	5.2 ± 8.9	6.4 ±10.8	0.570
FSF (mL)	148 ± 48.6	176 ± 110	133 ± 67.1	143 ± 94.3	0.140
FS (mL)	214 ± 117	249 ± 120	195 ± 96.0	235 ± 139	0.151
US (mL)	237 ± 125	280 ± 139	220 ± 110	281 ± 160	0.113
Compliance	24.2 ± 18.6	69.0 ± 82.2	40.5 ± 57.7	61.6 ± 82.1	0.147
Pdet (cmH_2_O)	38.5 ± 32.6	33.7 ± 20.5	35.8 ± 23.7	24.4 ± 17.1	0.038
Qmax (mL/s)	2.75 ± 5.26	3.64 ± 4.59	3.83 ± 3.96	2.78 ± 4.05	0.775
Volume (mL)	136 ± 166	73.3 ± 144	60.5 ± 80.6	31.8 ± 50.4	0.084
PVR (mL)	232 ± 153	281 ± 216	218 ± 148	285 ± 195	0.316
VE (%)	32.6 ± 41.1	25.1 ± 31.6	25.6 ± 29.5	16.1 ± 25.1	0.570
BCI	52.3 ± 43.4	42.0 ± 30.6	54.9 ± 29.0	38.3 ± 31.0	0.115
BOOI	33.0 ± 33.5	16.5 ± 22.5	28.1 ± 25.8	18.9 ± 16.0	0.074

FSF: first sensation of filling, FS: full sensation, US: urge sensation, Pdet: detrusor pressure, Qmax: maximum flow rate, PVR: post-void residual, VE: voiding efficiency, BC: bladder contractility index, BOOI: bladder outlet obstruction index. *p* values indicate the statistical analysis of the urodynamic variables among SCI patients who had different treatment outcome.

**Table 3 toxins-14-00030-t003:** Improvement in voiding dysfunction with surgical procedures and bladder management after the initial BoNT-A injections into the urethral sphincter.

	N	Not Improved	Mildly Improved	Moderately Improved	Markedly Improved	*p* Value
Self-voiding and on medication	55	4 (7.3%)	16 (29.1%)	28 (50.9%)	7 (12.7%)	0.472
Urethral BoNT-A	42	16 (38.1%)	14 (33.3%)	10 (23.8%)	2 (4.8%)	0.915
Detrusor BoNT-A/CIC	20	1 (5.0%)	6 (30.0%)	10 (50.0%)	3 (15.0%)	0.628
AE and CIC	5	0	2 (40%)	1 (20.0%)	2 (40%)	1.000
TUI-BN	16	1 (6.3%)	10 (62.5%)	4 (25.0%)	1 (6.3%)	0.011
TUI-P/TUR-P	9	0	3 (33.3%)	4 (44.4%)	2 (22.2%)	1.000
Suburethral sling	2	0	1 (50.0%)	1 (50.0%)	0	1.000
Cystostomy	2	0	1 (25.0%)	3 (75.0%)	0	0.157
Indwelling catheter	4	0	3 (75.0%)	1 (25.0%)	0	1.000
CIC alone	10	1 (10.0%)	4 (40.0%)	5 (50.0%)	0	0.517
External sphincterotomy	5	2 (40.0%)	1 (20.0%)	1 (20.0%)	1 (20.0%)	0.386

Patients might have >1 bladder management or surgical procedure after initial urethral BoNT-A injection; BoNT-A: botulinum toxin A, CIC: clean intermittent catheterization, AE: augmentation enterocystoplasty, TUI-BN: transurethral incision of bladder neck, TUI-P: transurethral incision of the prostate, TUR-P: transurethral resection of the prostate. *p* value: statistics of each bladder management or surgical procedure between not improved plus mildly improved versus moderately and markedly improved.

**Table 4 toxins-14-00030-t004:** Overall satisfaction to the current bladder and voiding conditions with bladder management and surgical procedures after the initial BoNT-A injection into the urethral sphincter.

	N	Very Satisfied	Satisfied but Wish to Change	Unsatisfied but No Change	Very Unsatisfied	*p* Value *
	118	40 (33.9%)	30 (25.4%)	41 (41.7%)	7 (5.9%)	
**Level of SCI** **Cervical** **Thoracic** **Lumbo-sacral**	51 43 24	17 (33.3%) 17 (39.5%) 6 (25.0%)	13 (25.5%) 12 (27.9%) 5 (20.8%)	18 (35.3%) 13 (30.2%) 10 (41.7%)	3 (5.9%) 1 (2.3%) 3 (12.5%)	0.630
**Self-voiding and** **on medication**	55	16 (29.1%)	12 (21.8%)	23 (41.8%)	7 (7.3%)	0.395
**Urethral BoNT-A**	42	16 (38.1%)	14 (33.3%)	10 (23.8%)	2 (4.8%)	0.228
**Detrusor BoNT-A/CIC**	20	7 (35.0%)	7 (35.0%)	5 (25.0%)	1 (5.0%)	0.666
**AE and CIC**	5	4 (80.0%)	0	1 (20.0%)	0	0.249
**TUI-BN**	16	10 (62.5%)	4 (25.0%)	2 (12.5%)	0	0.051
**TUI-P/ TUR-P**	9	2 (22.2%)	4 (44.4%)	3 (33.3%)	0	0.560
**Suburethral sling**	2	2 (100%)	0	0	0	0.292
**Cystostomy**	2	0	0	2 (100%)	0	0.410
**Indwelling catheter**	4	0	3 (75.0%)	1 (25.0%)	0	0.165
**CIC alone**	10	5 (50.0%)	1 (10.0%)	3 (30.0%)	1 (10.0%)	0.394
**External sphincterotomy**	5	2 (40.0%)	1 (20.0%)	2 (40.0%)	0	1.000

* *p* value: comparison of the rate of patients with very satisfied and satisfied but wish to change among different subgroups of SCI level, and between the patients with and without bladder management or surgical procedure, Patients might have >1 bladder management or surgical procedure after initial urethral BoNT-A injection; BoNT-A: botulinum toxin A, CIC: clean intermittent catheterization, AE: augmentation enterocystoplasty, TUI-BN: transurethral incision of bladder neck, TUI-P: transurethral incision of the prostate, TUR-P: transurethral resection of the prostate.

## Data Availability

Data is available on request to the corresponding author.

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
