# Peer review of "Real-World Data Regarding Satisfaction to Botulinum Toxin A Injection into the Urethral Sphincter and Further Bladder Management for Voiding Dysfunction among Patients with Spinal Cord Injury and Voiding Dysfunction"

_toxins, 2022, doi:10.3390/toxins14010030_

Round 1
Reviewer 1 Report
In this article, authors report interesting results on a retrospective ten-year follow-up study of LM patients treated (or not) with botulinum neurotoxin injection into the urethral sphincter and further bladder management for voiding dysfunction. Overall, the retrospective study is well conducted and its interpretation is appropriate. The results are very interesting, complete and very detailed. I am impressed that, as stated by the authors, "the rate of improvement in voiding dysfunction after repeated injections of BoNT-A into the urethral sphincter was less favorable than that after surgery or medical treatment." This means that treatment with BoNT-A in patients with SCI, having positive effects in the short term, is perhaps not as favorable in a long-term treatment with BoNT-A. I have no major concerns or criticisms about this report and I think the manuscript may be accepted for publication on Toxins after a few small corrections below.
Minor
General comment: perhaps it would be useful to report (in materials and methods) whether the group of patients analyzed in the retrospective study were from the same hospital, treated by the same medical team, as it probably is. If not, I wonder if the authors observed whether there might be differences within the groups due to "environmental" differences. If there were no environmental differences, a sentence in discussion that excludes them is sufficient.
Line 93: …Bonita (?) do you mean “BoNT-A”?
Line 162: …Bonet-A (?) do you mean “BoNT-A”?
Author Response
Reviewer #1
In this article, authors report interesting results on a retrospective ten-year follow-up study of LM patients treated (or not) with botulinum neurotoxin injection into the urethral sphincter and further bladder management for voiding dysfunction. Overall, the retrospective study is well conducted and its interpretation is appropriate. The results are very interesting, complete and very detailed. I am impressed that, as stated by the authors, "the rate of improvement in voiding dysfunction after repeated injections of BoNT-A into the urethral sphincter was less favorable than that after surgery or medical treatment." This means that treatment with BoNT-A in patients with SCI, having positive effects in the short term, is perhaps not as favorable in a long-term treatment with BoNT-A. I have no major concerns or criticisms about this report and I think the manuscript may be accepted for publication on Toxins after a few small corrections below.
Minor
General comment: perhaps it would be useful to report (in materials and methods) whether the group of patients analyzed in the retrospective study were from the same hospital, treated by the same medical team, as it probably is. If not, I wonder if the authors observed whether there might be differences within the groups due to "environmental" differences. If there were no environmental differences, a sentence in discussion that excludes them is sufficient.
Reply: Thank you for the comments. The data of patients in this study were collected from one medical center and all treatments were performed by the same urology team.
Line 93: …Bonita (?) do you mean “BoNT-A”?
Reply: Thank you for the correction. We have revised it to BoNT-A.
Line 162: …Bonet-A (?) do you mean “BoNT-A”?
Reply: Thank you for the correction. We have revised it to BoNT-A.
Reviewer 2 Report
The manuscript is well written and worthy of publication in Toxins. In the text, there are some typos in indicating the BoNT-A (e.i., line 72 “BoN-A”; line 93 “Bonita”; line 162 “Bonet-A”; line 244 BoN-A”), but I think that these typos can be rectified during the proof correction.
Since Toxin is an open-access journal and there are no limitations in the number of pages for the paper, I would suggest that the authors consider the opportunity of limiting the number of acronyms to make it easier to read the text. For example, the phrase from line 42 to line 45 is hard to read.
Author Response
Reviewer #2
The manuscript is well written and worthy of publication in Toxins. In the text, there are some typos in indicating the BoNT-A (e.i., line 72 “BoN-A”; line 93 “Bonita”; line 162 “Bonet-A”; line 244 BoN-A”), but I think that these typos can be rectified during the proof correction.
Reply: Thank you for the correction. We have revised all the typos to BoNT-A.
Since Toxin is an open-access journal and there are no limitations in the number of pages for the paper, I would suggest that the authors consider the opportunity of limiting the number of acronyms to make it easier to read the text. For example, the phrase from line 42 to line 45 is hard to read.
Reply: Thank you for the correction. We have reduced the number of abbreviations to 10 in order to make the manuscript easier to read.
Reviewer 3 Report
This manuscript described a small retrospective study using botulinum toxin A injection into urethral sphincter for management for voiding dysfunction among patients with chronic spinal cord injury. The results suggested that the therapeutic efficacy of BoNT-A injection into the urethral sphincter for voiding dysfunction in patients with chronic spinal cord injury is limited, and the repeated injection for voiding dysfunction is less favorable than other treatments (including surgery). While the patient population is small, it is useful to see the limited efficacy and less acceptable among patients with chronic spinal cord injury using BoNT-A injection into the urethral sphincter for voiding dysfunction.
However, the authors need to clarify their statistical analysis results. The main issue is the p-value in all tables (only Table 3 specified p-value meanings). The authors need to explain and interpret the statistical analysis. For example, in Table 1, the p-value is for comparing “Not improved and mildly improved” with “moderately improved and markedly improved”? If so, do they use the pare-wise analysis, or combine the different subgroups together?
The author claimed that there is a significant improvement in voiding conditions after BoNT-A injection, however, from Table 1, none of the p-values are less than 0.05.
Line 93, “Bonita injection” seems as a typo (BoNT-A infection?)
Table 3, P value# P value redundant.
Author Response
Reviewer #3
This manuscript described a small retrospective study using botulinum toxin A injection into urethral sphincter for management for voiding dysfunction among patients with chronic spinal cord injury. The results suggested that the therapeutic efficacy of BoNT-A injection into the urethral sphincter for voiding dysfunction in patients with chronic spinal cord injury is limited, and the repeated injection for voiding dysfunction is less favorable than other treatments (including surgery). While the patient population is small, it is useful to see the limited efficacy and less acceptable among patients with chronic spinal cord injury using BoNT-A injection into the urethral sphincter for voiding dysfunction.
However, the authors need to clarify their statistical analysis results. The main issue is the p-value in all tables (only Table 3 specified p-value meanings). The authors need to explain and interpret the statistical analysis. For example, in Table 1, the p-value is for comparing “Not improved and mildly improved” with “moderately improved and markedly improved”? If so, do they use the pare-wise analysis, or combine the different subgroups together?
Reply: Thank you for the comment. We have added the description of the statistical analysis in the Method section, and also in the footnote of each table.
The author claimed that there is a significant improvement in voiding conditions after BoNT-A injection, however, from Table 1, none of the p-values are less than 0.05.
Reply: Thank you for the comment. We have revised the statement as following: After BoNT-A injections into the urethral sphincter, 71 (60.2%) patients, including 18 (15.3%) with markedly and 53 (44.9%) with moderate improvement, had improvement in voiding condition.
Line 93, “Bonita injection” seems as a typo (BoNT-A infection?)
Reply: Thank you for the correction. We have revised the typo to BoNT-A.
Table 3, P value# P value redundant.
Reply: Thank you for the correction. We have deleted the P value # and related data presentation in Table 3.
Round 2
Reviewer 3 Report
There are still some statistic analysis need to e clarified in Tables 1 and 4, regarding the subgroups For example, for both tables, under level of SCI, is the p-value is based on combined of Cervical, Thoracic and Lumbo-sacral? And for Table 1, is the p-value under NDO - DSD for the combined NDO - DSD, NDO+AD+DSD, NDO+DSD, DA+DSD, and DA+ ISD? Same for other p-values in Table 1.
This will not change the conclusion since there is no significant improvement on patients' satisfaction using BoNT/A comparing with other bladder management.
In the abstract, authors claimed that "Meanwhile, others..... had significant improvement in voiding 25 condition and overall satisfaction." But Table 3 did not showed significant changed (p>0.05). Authors need to clarify/correct that statement and associated contents in the text.
Author Response
Reviewer #3
There are still some statistic analysis need to e clarified in Tables 1 and 4, regarding the subgroups For example, for both tables, under level of SCI, is the p-value is based on combined of Cervical, Thoracic and Lumbo-sacral? And for Table 1, is the p-value under NDO - DSD for the combined NDO - DSD, NDO+AD+DSD, NDO+DSD, DA+DSD, and DA+ ISD? Same for other p-values in Table 1.
This will not change the conclusion since there is no significant improvement on patients' satisfaction using BoNT/A comparing with other bladder management.
Reply: Thank you for the comment. The statistics in tables have been described in the methods section (lines 356-361) We have added definition of p values in the footnote of the table 1 (lines 105-106) and table 4 (lines 162-164); and also added statements in the result section. (lines 98-101)
In the abstract, authors claimed that "Meanwhile, others..... had significant improvement in voiding 25 condition and overall satisfaction." But Table 3 did not showed significant changed (p>0.05). Authors need to clarify/correct that statement and associated contents in the text.
Reply: Thank you for the comment. Wee have revised the statement in abstract: Meanwhile, more than 60% of patients…… had moderate and marked improvement in voiding dysfunction and overall satisfaction. (lines 23-26)